# Multivariate genome-wide association study on tissue-sensitive diffusion metrics highlights pathways that shape the human brain

Chun Chieh Fan [1,2,3 ✉], Robert Loughnan[4], Carolina Makowski[2,3], Diliana Pecheva[2,3], Chi-Hua Chen[3], Donald J. Hagler Jr.[2,3], Wesley K. Thompson[1,3], Nadine Parker[5], Dennis van der Meer[5,6], Oleksandr Frei[5,7], Ole A. Andreassen [5] & Anders M. Dale [2,3,4,8]

The molecular determinants of tissue composition of the human brain remain largely unknown. Recent genome-wide association studies (GWAS) on this topic have had limited success due to methodological constraints. Here, we apply advanced whole-brain analyses on multi-shell diffusion imaging data and multivariate GWAS to two large scale imaging genetic datasets (UK Biobank and the Adolescent Brain Cognitive Development study) to identify and validate genetic association signals. We discover 503 unique genetic loci that have impact on multiple regions of human brain. Among them, more than 79% are validated in either of two large-scale independent imaging datasets. Key molecular pathways involved in axonal growth, astrocyte-mediated neuroinflammation, and synaptogenesis during development are found to significantly impact the measured variations in tissue-specific imaging features. Our results shed new light on the biological determinants of brain tissue composition and their potential overlap with the genetic basis of neuropsychiatric disorders.

[1] Population Neuroscience and Genetics Lab, University of California, San Diego, La Jolla, CA, USA. [2] Center for Multimodal Imaging and Genetics, University of California, San Diego, La Jolla, CA, USA. [3] Department of Radiology, School of Medicine, University of California, San Diego, La Jolla, CA, USA. [4] Department of Cognitive Science, University of California, San Diego, La Jolla, CA, USA. [5] NORMENT Centre, Division of Mental Health and Addiction, Oslo University Hospital & Institute of Clinical Medicine, University of Oslo, Oslo, Norway. [6] School of Mental Health and Neuroscience, Faculty of Health, Medicine and Life Sciences, Maastricht University, Maastricht, Netherlands. [7] Centre for Bioinformatics, Department of Informatics, University of Oslo, Oslo, Norway. [8] Department of Neuroscience, University of California, San Diego, La Jolla, CA, USA. ✉email: c9fan@ucsd.edu

The human brain develops through complex yet carefully orchestrated neurobiological processes, whereby cortical and subcortical circuitries are integrated for proper functioning[1]. Neural migration, axonal guidance, and synapse formation are coordinated through spatially distributed molecular gradients spanning across several brain regions[2]. Differences in tissue composition are the result of these developmental processes. We can gain substantial insight into how neural circuitries were formed and supported by investigating the genetic determinants of whole-brain patterning with respect to tissue composition.

Recent advances in multi-shell diffusion magnetic resonance imaging and diffusion signal modeling have created an opportunity to evaluate tissue composition in vivo[3–7]. Differences in signals between water molecules of intracellular, extracellular, and unhindered compartments are captured by higher-order diffusivity (multiple shells), allowing for the estimation of the relative proportions of cell bodies, axonal fibers, and interstitial fluids within a voxel[3–5,8–12]. This type of tissue modeling has been used to detect compositional changes driven by neurodegeneration[8,11], development[3], obesity[4], and carcinogenesis[9,10]. However, there is currently no genome-wide association study (GWAS) on compositional features. This omission is critical, as traditional imaging measurements are insensitive to neurite density, short-range fibers, and cellular properties of cortical gray matter and subcortical nuclei[7].

Moreover, GWAS of brain imaging measurements usually adopt a univariate approach, performing associations with one brain region at a time[13–18]. Patterns encompassing the whole brain have been mostly ignored or controlled away as global effects, potentially biasing the interpretations toward purely regional effects. This risks misattributing the nature of genetic effects on the brain, e.g., the cortical surface area is driven by local cortical expansion when it may instead be due to underlying axonal growth. The univariate region-of-interest approach may also be underpowered to detect the full extent of genetic variants associated with canonical neurodevelopmental pathways, especially when effects are spatially distributed[19,20]. A multivariate GWAS, focused on detecting loci that have effects across multiple brain regions, has been shown to be highly efficient in discovering many loci[21–23]. By expanding how we can analyze the discovered spatial patterning from the multivariate GWAS, we can reveal further biological insight into the molecular gradients that shape the human brain.

Here, we performed a multivariate GWAS on the metrics derived from multi-shell diffusion imaging to examine the genetic determinants of whole-brain patterning of cellular compartments. Using two largest extant imaging genetic studies that have compatible multishell scans, the UK Biobank[24] (UKB) and the Adolescent Brain Cognitive Development[SM] Study (ABCD Study[®])[25,26], we identified and validated 503 unique loci for tissue sensitive diffusion metrics. The discovered loci were enriched for neurogenesis, neuron differentiation, and axonal development. Among the validated loci, 152 have not been reported previously by GWAS of brain imaging phenotypes. By investigating the spatial distribution of the associated effects, we highlighted critical molecular pathways involved in neuroinflammation and axonal growth, and the corresponding regions that may be susceptible to these processes. Signal overlap, at both the locus level and genome-wide, with neuropsychiatric outcomes indicate the functional relevance of our GWAS results, providing a foundation for further understanding of the biological underpinnings of neuropsychiatric disorders.

## Results

### Multivariate GWAS on features of tissue composition across the whole brain.
We processed multi-shell diffusion MRI data from UKB and ABCD with restriction spectrum imaging (RSI) to extract the tissue composition features of the human brain[3–5,8–12,27]. To ensure the validating test was robust against study variability due to time shift[28], we selected the UKB samples received MRI scans before 2019 as the discovery set while all others were regarded as replication sets. The sample characteristics can be found in the Supplementary Table 1. The images were harmonized and registered to a common atlas to ensure the alignment of voxels across subjects (See Method for detailed imaging processing pipelines[25,26] and Supplementary Figure 1 for quality control metircs). RSI decomposes the diffusion-weighted signals as emanating from three separable tissue compartments: intracellular, extracellular, and free water (Fig. 1a). Each compartment is characterized by its intrinsic diffusion properties. In this study, we consider the intracellular compartment, which is defined by restricted diffusion bounded by cellular membranes, and the free water compartment characterized by the unimpeded diffusion of water molecules. RSI estimates the normalized isotropic restricted signal volume fraction, N0, which captures the relative amount of cell bodies within a voxel, such as the densities of neurons, astrocytes, and oligodendrocytes. The normalized directional restricted signal volume fraction, ND, captures the relative amount of tube-like structures within a voxel, such as axons and dendrites. The free water component, NF, captures the relative amount of free water outside of cell structures. N0, ND, and NF provide greater tissue specificity than the widely-used diffusion tensor metrics, have been useful in the understanding variation of cellular organization within the human brain and are highly informative for the human brain development[3,10,11,27]. The spatial distributions of those three tissue-sensitive measures can be seen in Supplementary Figure 2–5.

Three separate voxel-wise multivariate GWAS on N0, ND, and NF were performed. For the discovery stage (UKB discovery set, imaging acquisition before 2019, $n = 23,543$), we used combined principal components (CPC) statistics[22,29] (Fig. 1b) to identify associated loci from multivariate measurements. As a practical extension to other multivarite GWAS methods, such as MOSTest[21], CPC combines statistics from associations with the finite number of principal components and has close form expression on the null distribution without the need for permutations[22]. Using the UKB discovery set, we calculated the principal components (PCs) from the tissue feature across all voxels. From the whole-brain images in 2 mm resolution per voxel, spanning across 100 by 100 by 130 voxels, the first 5000 PCs were extracted and used in the subsequent analyses, explaining more than 70% of the total variance of the imaging data (Supplementary Figure 6). Since all PCs are orthogonal to each other, the statistical inference can be based on combining the associations between genetic variants and each of the derived PCs (Fig. 1b). Each of the PCs can be regarded as an orthogonal basis function with limited interpretability, yet the weighted combination of them can represent any spatial distribution (Supplementary Figure 7–9). CPC combined the association signals across PC for a given genetic variant and detect the genetic loci that are shared across multiple PCs, thus reducing the burden of multiple testing and the false detection on nuisance effects. We tuned the hyper-parameters for the combination function to optimize the power for discovery[19,22] by searching through four possible combination sets (see Methods). To account for hyper-parameter tuning and the three tissue features, we set the p-value threshold for genome-wide significance as 5e-8 divided by 12 = 4.2e-9.

After Linkage-disequilibrium pruning (LD $R^2 > 0.1$) and positional clumping (distance < 250 K bp), we found 432, 350, and 273 independent genetic loci associated with N0, ND, and NF, respectively (Fig. 2a; Supplementary Data 1–4). After merging loci with overlapping genomic ranges, there are 503

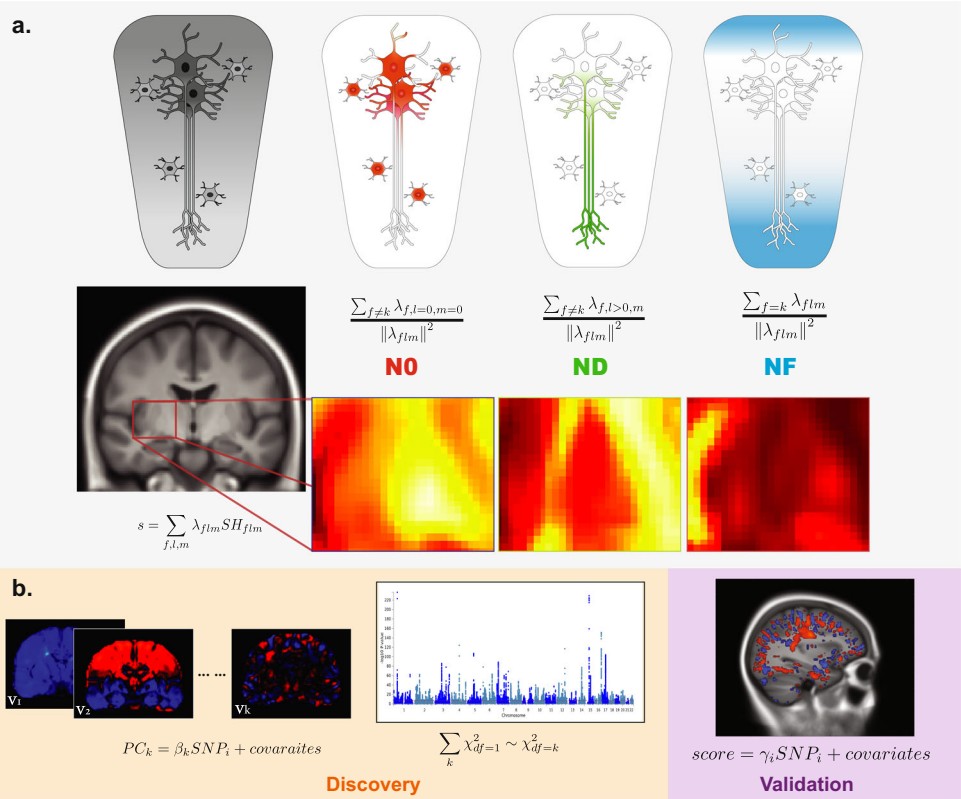

**Fig. 1 Overview of the study design. a** Illustration of tissue composition imaging features. The first row highlights which cellular compartments the metrics intend to capture. The second row is the formula used for calculating each metric, i.e. N0, ND, and NF (see Methods section). The third row shows the actual signal intensities for N0, ND, and NF, respectively. **b** Illustration of the analytic sequence of multivariate GWAS. The discovery stage involves summarizing the whole-brain voxel-wise data into k principal components (PCs) and then performing the GWAS inference based on combined association signals across PCs (UKB discovery, $n = 23,543$). The validation stage involves confirmatory associations with polyvoxel scores (UKB replication, $n = 6396$; ABCD, $n = 8189$).

unique loci across all three tissue features (Supplementary Data 1).

**Loci validated in adults and adolescents**. To validate the discovered loci in independent studies, we first calculated polyvoxel scores[30–33] based on eigenvectors and association weights from the discovery set, and then performed the association tests between genetic variants and the derived scores (see Methods). This procedure is similar to confirmatory canonical correlation analysis[23], except with only one variant involved in each regression. We repeated the same confirmatory analysis in the UKB validation set ($n = 6396$, scanned after 2019) and ABCD samples ($n = 8189$), except for including study-specific covariates and random effects controlling for family relatedness and diverse genetic background in ABCD (see Methods). Among the discovered loci, 335 (79%), 298 (85%), and 222 (81%) were found to be validated in the independent UKB validation set for N0, ND, and NF, after Bonferroni correction for the number of loci discovered. In ABCD, 106 (25%), 153 (43%), and 88 (32%) of the discovered loci were validated for N0, ND, and NF, despite the large differences in age and other sample characteristics between UKB and ABCD.

**Characteristics of validated loci**. To examine the overlap between our validated loci and previously reported loci in neuroimaging GWAS, we curated the reported loci lists from the NHGRI-EBI Catalog based on keywords in "brain", "imaging", "cortical", "subcortical", and "white matter". The final list of reported loci included GWAS on brain connectivity[15], cortical surface

measures[13,21,34], derived imaging instruments across all modalities[35], subcortical volumes[14,21,36], brain volumes[16,34,37], white matter hyperintensities[38], and white matter microstructure[18]. We queried if any of our validated loci were in linkage-disequilibrium (LD) with or located in 250 kb regions of previously reported neuroimaging loci. The results are summarized in Fig. 2b. Among the validated loci, 134 unique loci overlapped with previously published GWAS on cortical surface measurements and 108 unique loci were found to be associated across cortical and subcortical structures, indicating wide pleiotropic effects across brain regions (Fig. 2b, Supplementary Data 2–7). We also found 136 unique novel loci through our approach, demonstrating improved power in both discovery and replications.

On the other hand, the gene set analyses[39,40] on the identified loci shows each tissue feature has distinct pattern of Gene Ontology enrichment. While all tissue features were highly enriched for the Gene Ontology term of neurogenesis, N0 showed stronger enrichment in anatomical morphogenesis, while ND demonstrated more enriched in axon development, neuron projection guidance, and tangential neuronal migration (Fig. 2c). This suggests that at the level of the genomic loci, modeling tissue compositions captured differential molecular effects associated with the human brain.

**Loci showing differential effects on tissue compositions**. Closer inspection of the effect size distributions of the loci provides a unique angle into the molecular processes shaping the human brain. For instance, the 5q14.3 locus at the gene body of *VCAN*, tagged by a common SNP rs12653308, was found to be strongly associated

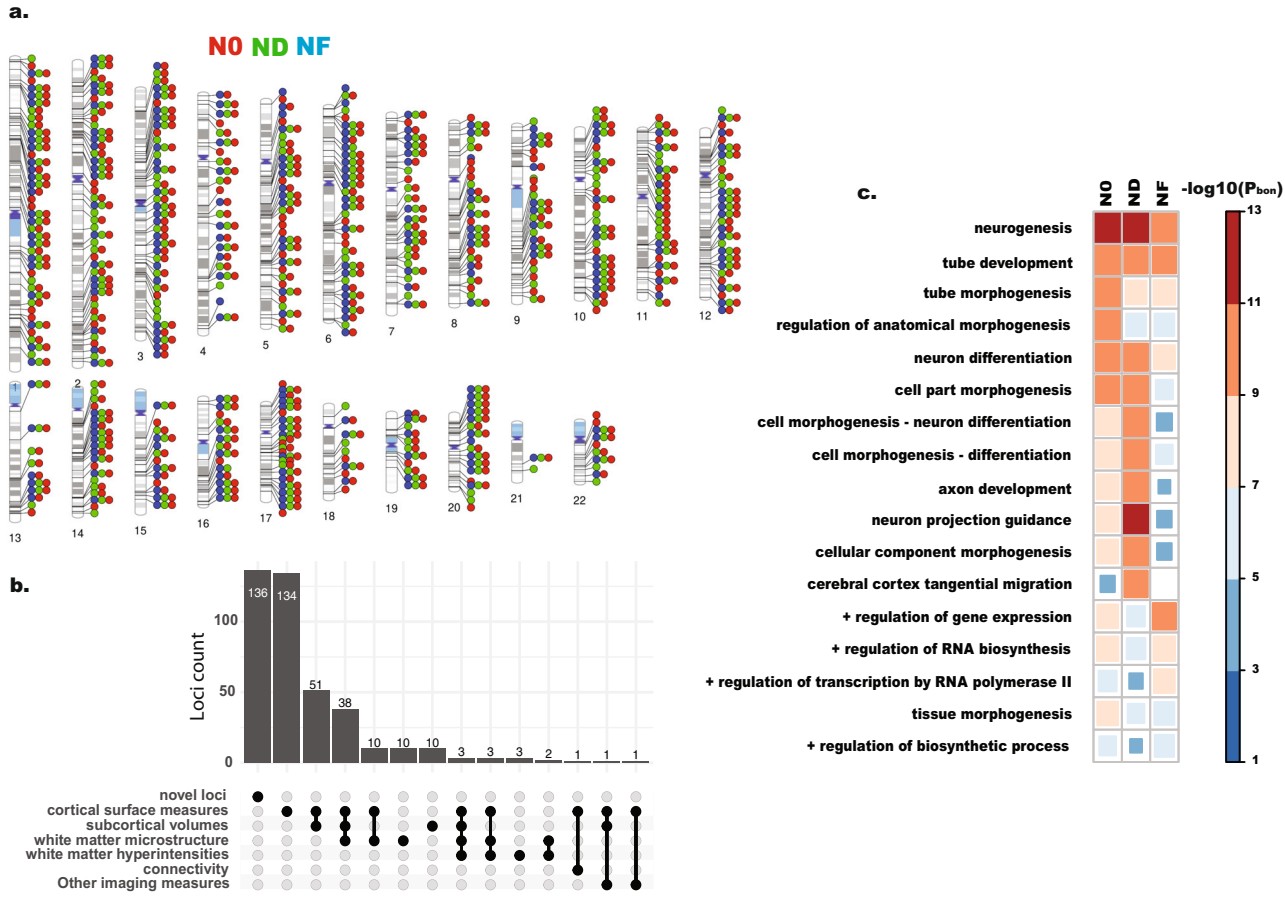

**Fig. 2 Results of multivariate GWAS on whole-brain imaging features. a** Ideogram of the discovered loci, colored according to the imaging features. **b** Offset plot shows the unique and overlaping validated loci with previous neuroimaging GWAS, including brain connectivity[15], cortical surface measures[13,21,33], derived imaging instruments across all modalities[34], subcortical volumes[14,16,21,33,35,36], white matter hyperintensities[37], and white matter microstructures[18]. **c** The top-ranking enriched Gene Ontologies, derived from gene set analyses on significant loci from each imaging feature. +: positive regulation. All p values were bonferroni corrected, based on the two-sided tests with gene set analysis. Results all based on the GWAS summary statistics from the UKB discovery (n = 23,543).

with N0 (Fig. 3a). It was reported to be associated with various diffusion metrics from white matter fiber tracts[18] and cortical surface measurements[21] (Fig. 3a). Instead of fiber tracts or cortical surface regions, we found that the association strength is particularly strong in the hippocampus bilaterally (Figs. 3b, c, Supplementary Data 5–8), based on the regional enrichment analysis with 50,000 bootstraps (see Method). *VCAN*, which encodes versican and is a lectican-binding chondroitin sulfate proteoglycan (CSPG), serves a critical role in astrocyte-mediated neuroinflammation[41], and has potential interacting pharmacological targets[42,43] (Fig. 3d; Supplementary Information; Supplementary Data 9–11). CSPGs were found to be associated with astrocyte-dependent synaptogenesis within the hippocampus[44]. When we examined the associations between genetic variants of genes encoding CSPGs (*BCAN*, *NCAN*, and *VCAN*) and tissue features, we found N0 showed stronger association signals than ND and NF (Fig. 3e). Since the effects were validated in ABCD, our results support the early effects of astrocytic mediated processes on the human hippocampus via CSPGs. Changes in the distribution of CSPGs in the hippocampal formation were observed among patients with schizophrenia and patients with bipolar disorders[45,46], linking our findings to neuropsychiatric outcomes.

The locus located at 2p23.3, tagged by rs11126784, has strong signals associated with ND (Fig. 3f). This locus resides within the

gene body of *DPYSL5* and has been reported to be associated with cortical surface measures[21]. Instead of the cortical surface, our whole-brain multivariate GWAS indicates the effect sizes were more diffusely distributed among white matter tracts, especially within cortico-striatal circuitry (Fig. 3g, h, Supplementary Data 5-8). *DPYSL5* belongs to the collapsin response mediator protein (CRMP) family, including *DPYSL2*, *DPYSL3*, and *DPYSL4*, which are essential for axonal growth and neurite morphogenesis[47–49] (Fig. 3i). Indeed, all tagged SNPs of the CRMP family proteins show stronger association signals with ND than with N0 and NF (Fig. 3j). Our results are concordant with CRMP involvement in neurodevelopment and showing that their effects can be observable among major white matter fiber bundles early on. Our findings are also relevant to neuropsychiatric outcomes, as CRMP has been implicated in schizophrenia and mood disorders[50].

The 136 novel loci we discovered and validated in this study are relevant for neuropsychiatric phenotypes and warrant further investigation (Supplementary Data 2-4). An N0-specific novel locus at 5q14.3 is within the gene body of *MEF2C*, which can influence neural progenitor cell differentiation and regulation of synaptic densities[51,52]. This locus overlaps with GWAS findings of educational attainment and intelligence[53]. Another locus at 20p12.1, on the gene body of *MACROD2*, showed consistent signals among adults and adolescents (ND: UKB discovery

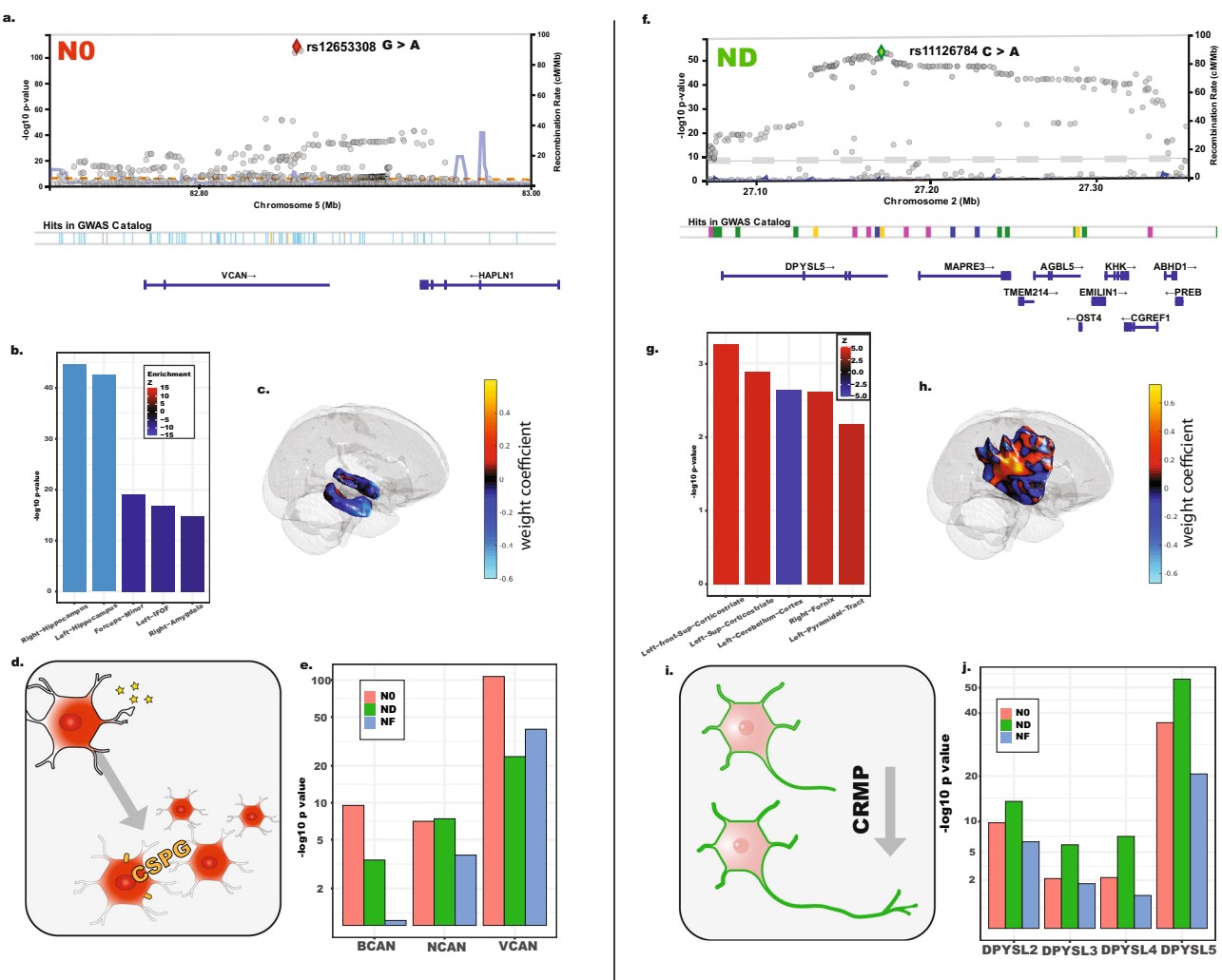

**Fig. 3 Illustrations of the selected loci. a** Locus plot of the association signals with N0 in 5q14.3. **b** Regional enrichment of the association signals on the human brain. The top five enriched regions are shown. **c** 3D visualization of the top two enriched regions (hippocampus). **d** Visual representations of the functions of CSPG. **e** Association magnitudes across CSPGs, including *BCAN*, *NCAN*, and *VCAN*. (UKB discovery, $n = 23{,}543$) **f** Locus plot of the association signals with N0 in 2p23.3. **g** Regional enrichment of the association signals on the human brain. The top five enriched regions are shown. **h** 3D visualization of the top two enriched regions (cortical striatum). **i** Visual representations of the functions of CRMP. **j** Association magnitudes across CRMP, including *DPYSL2*, *DPYSL3*, *DPYSL4*, and *DPYSL5*. (UKB discovery, $n = 23{,}543$).

$p = 1\mathrm{e}{-}29$, UKB validation $p = 1.7\mathrm{e}{-}18$, and ABCD validation $p = 4.6\mathrm{e}{-}8$, and has previously been linked to autism[54] and general cognitive ability[55]. The gene *MACROD2* was also implicated in educational attainment[53] and risk-taking behaviors[56].

**Cell-type enrichment analysis**. Although N0, ND, and NF were designed to capture different properties of tissue compartments, the strong overlapping signals across the three features indicates that similar cell processes and populations may shape all three microstructural features. To investigate this, we analyzed the heritability enrichment given cell type annotations using stratified LD score regression (S-LDSC)[57]. A dimensionally-corrected multivariate statistic, such as the scaled $\chi^2$, can be used in the context of LDSC for deriving the relative enrichment in the average heritability of the high-dimensional phenotypes[23]. Hence, we ran S-LDSC with tissue-specific chromatin annotations[57] and cell type-specific annotations[58] to obtain cell type-specific enrichment patterns for our RSI phenotypes (Fig. 4).

While the overall patterns of the enrichment are similar across three tissue features, ND has the strongest enrichment signals across all activating histone markers (H3K27ac, H3K36me3, H3K4me1, H3K4me3, and H3K9ac) and DNase hypersensitivity sites ($P_{bon} < 0.05$). All three features were enriched in the chromatin state of fetal brain and hippocampal tissues whereas ND also shows enrichment in the cingulate cortex and substantia nigra (Fig. 4a). With respect to cell populations, using public available cell-type-specific chromatin state data from mouse samples that have been shown to be useful for prioritizing human GWAS results[58], our analysis indicates all three features have significant enrichment in embryonic dopaminergic interneurons and astrocytes ($P_{bon} < 0.05$; Fig. 4b). Moreover, ND shows stronger enrichment signals in oligodendrocytes, as expected for an imaging feature capturing the integrity of tubular structures such as the myelin sheath.

**Genetic overlap with neuropsychiatric and immune-related phenotypes**. We investigated the proportion of genome-wide signals of the three tissue features which overlap with neuropsychiatric phenotypes[53,56,59–65] and immune disorders[66].

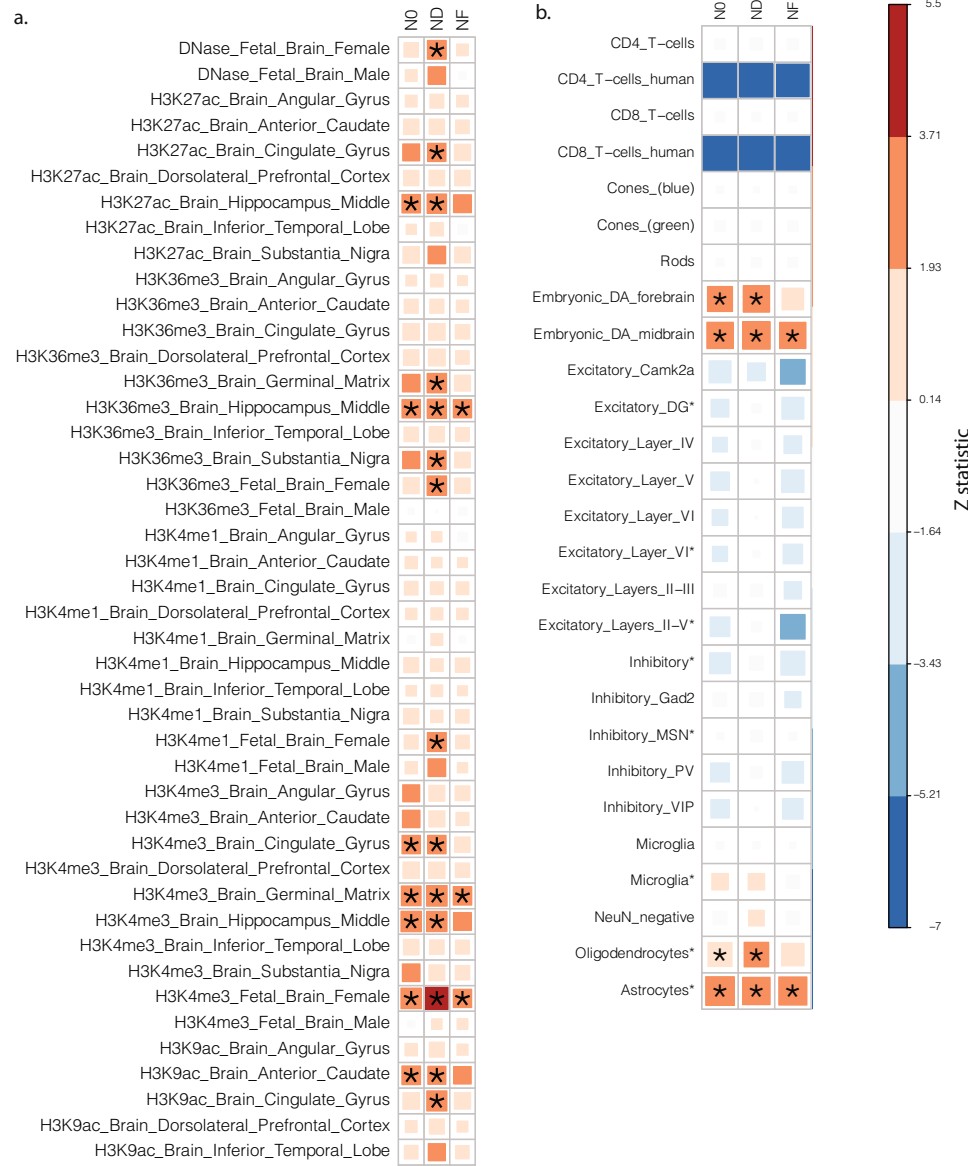

**Fig. 4 Cell-type specific enrichment results.** Results from stratified LDSC analysis with dimensionally corrected effect sizes. **a** Tissue-specific histone markers. **b** Cell types.

Based on a method tailored for unsigned multivariate statistics[23], we evaluated the signal shared between each pair of traits using their summary statistics. The amount of shared signal was the Spearman correlation of the average SNP $-\log_{10} p$ values within each approximately independent LD block. All three tissue features consistently show significant overlap with immune disorders ($\rho$:0.15–0.21, all $P < 1.2\text{e-9}$)[66], schizophrenia ($\rho$: 0.15–0.17, all $P < 5\text{e-10}$)[63], attention deficit hyperactivity disorder ($\rho$: 0.11–0.12, all $P < 2\text{e-6}$)[67], bipolar disorder ($\rho$: 0.10–0.12, all $P < 6\text{e-6}$)[61], and cross-psychiatric-disorders ($\rho$: 0.14–0.16, all $P < 7\text{e-9}$)[59]. The overlaps with Alzheimer's disease is less evident ($\rho$: 0.07–0.10)[65]. Educational attainment[53] and risk-related behaviors[56] are also significantly correlated ($\rho$: 0.10–0.20, all $P < 5\text{e-5}$; Supplementary Figure 11). The patterns of genome-wide signals shared with neuropsychiatric phenotypes were not evidently different across the three tissue features, despite the distinct patterns we observed at the locus level and cell-type-specific enrichments. While the limited resolution of LD blocks may

contribute to this null finding, the evident similarities in the genome-wide level results may mean that pleiotropic effects, either horizontal or vertical, on neurodevelopmental traits are highly polygenic, sharing multiple loci but with different functional outputs.

## Discussion

Using imaging features of whole brain tissue compositions, a multivariate GWAS discovered and validated 503 loci, of which 136 had not been reported in previous GWAS of neuroimaging phenotypes. Through in-depth examination of effect size distributions, we demonstrated the specific impact of molecular pathways, including CSPGs and CRMP, on the tissue composition underlying the human brain in vivo. Our findings are relevant for neuropsychiatric outcomes, including cognitive functions and psychiatric disorders. By identifying the key protein families and highlighting the susceptible brain regions through

enrichment analyses, these results indicate a path to further investigate molecular mechanisms of brain regional development and specialization.

Our results indicate widespread pleiotropies between the development of cortical surfaces and cerebral white matter. As the patterning of the mature brain is the end results of multiple molecular processes working from differentiation of neuroprogenitor cells, migration of neurons, to synaptic prunings[68], the relevant genes are unlikely to confine their effects on one single anatomically defined region. This is in line with findings from malformations of cortical development that the germline mutations of genes involves in cell migration would lead to global malformations instead of localized lesions[69]. Many of the loci we discovered and replicated were also found to be associated with other imaging modalities across cortical and subcortical regions. We are not claiming that focal effects do not exist, indeed the variants we highlighted do have locally enriched signals. Instead we suggest that it is more likely that regional specification of the human brain is the ultimate result of a complex coordination of multiple distributed molecular processes[70] rather than that single genetic variants effect single anatomical regions. For instance, the group of genes belonging to CSPG had consistent associations with N0 metrics. The association signals were enriched in multiple brain regions beyond previously reported ROIs[18,21], especially bilateral hippocampus. This astrocyte-dependent molecular process may have more direct effects on the synaptic pruning in the hippocampal regions and then cascading downstream to the associated fiber tracts.

Our findings showcase the need for novel analytic approaches in brain imaging genetics. Multivariate GWAS on whole-brain phenotypes circumvents the potential "spotlight bias" that region-of-interest approaches are susceptible to[71]. Diffuse effects across brain regions and neurobiological pathways are more easily detected with this approach, as the inference is based on the total sum of the effects. Moving beyond the metrics of structural volumes or fiber orientation enabled us to detect molecular effects on brain tissue properties, identifying relevant biological pathways important for human brain development and neuropsychiatric outcomes.

Because our multivariate GWAS was optimized for detecting signals shared across PCs, the statistical power may be less than ideal for detecting extremely sparse genetic effects, i.e. limited to only one or two PCs[19,21,22]. Although it is possible to have regionally specific genetic effects, our approach will be less senstive to detect such effects since our PCs captured information across the whole brain and were anatomically agnostic. Instead of having one PC to represent one particular anatomical structure, it was the weighted combinations of several PCs that highlighted certain anatomical structures. This is the benefit of using a multivariate GWAS, as it implicitly picks up the patterning signals without pre-defining the region of interest. However, these statistical properties can also make it difficult to interpret which anatomical regions are most relevant for a given discovered loci. To facilitate the interpretation, we implemented regional enrichment analyses, examining which anatomical structures have higher average signals compared to other regions.

Our results highlight the pleiotropic nature of genes involved in synaptic pruning, neuroinflammation, and axonal growth. The microglia-related molecular processes were implicated in multiple brain regions across cortical and subcortical structures. The significant loci overlaps between tissue-sensitive imaging metrics and psychiatric disorders implicates the etiological mechanisms beyond the neuronal growth, such as microglia-mediated synaptic pruning. Our identified genes may aid in experimental studies investigating interventions for neuropsychiatric outcomes.

## Methods

**UK Biobank samples**. The inclusion criteria for the UKB sample were as follows: individuals who had valid consent at the time the analyses were performed (Dec 2020), were genetically inferred as having European ancestry, and completed the neuroimaging protocols. Among individuals who were included in the analyses, we further divided samples into two groups based on when the neuroimaging was performed (before or after 2019). We decided to use this naturally occurring temporal cut-point instead of randomized allotment of the groups because of best practice considerations[20,28,72,73], avoiding potential systematic biases driven by temporally related imaging confounds. In particular, the potential time shift of the study design can lead to over-optimistic evaluation on the generalizability if random data split instead of time split was used[28]. Given our purpose is to discover and validate the biologically relevant effects, we used a conservative approach by selecting a naturally occurring time point as the selection criteria for discovery and replication sets in UKB. Individuals who had valid imaging data before 2019 were assigned as the discovery set ($n = 23,543$) and those who had valid imaging data, not before, but after 2019 were assigned to the validation set ($n = 6396$). The demographic information of the final selected UKB samples can be found in the Supplementary Table 1. Data from UKB is obtained under accession number 27412.

**Adolescent Brain Cognitive Development study (ABCD) samples**. For validating of results, we selected the full baseline data of the ABCD Study from public data release 3.0 (NDA DOI: 10.15154/1524729). Since ABCD was designed to recruit individuals with the diverse ancestral background which reflect the racial/ethnic composition of the United States, we did not exclude individuals based on their genetic ancestries, using linear mixed-effects models to control for the family relatedness and heterogeneous ancestral background. We only excluded those who did not have valid imaging and genetic data from release 3.0, resulting in 8189 individuals in the analyses. The demographic characteristics of the ABCD samples can be found in the Supplementary Table 1.

**Imaging data processing**. Both UKB and ABCD have diffusion imaging protocols that were compatible for applying RSI models. The MRI scans of UKB were performed at three scanning sites in the United Kingdom, all on identically configured Siemens Skyra 3 T scanners, with 32-channel receive head coils. The MRI scans of ABCD were collected by 21 study sites throughout the United States, with scanners from Siemens Prisma, GE 750 and Phillips 3 T scanners. To harmonize the imaging data across the two studies, we processed the dMRI data from UKB and ABCD using the ABCD-consistent imaging processing pipeline implemented by the ABCD Data Analysis, Informatics, and Resource Center (ABCD DAIRC). The detailed processing procedures have been published elsewhere[25]. In short, multi-shell diffusion MRI data of ABCD acquired with seven $b = 0$ s/mm² frames and 96 noncollinear gradient directions, with 6 directions at $b = 500$ s/mm², 15 directions at $b = 1000$ s/mm², 15 directions at $b = 2000$ s/mm², and 60 directions at $b = 3000$ s/mm². Multishell diffusion MRI data of UKB acquired with five $b = 0$ s/mm² frames and 100 non-collinear gradient directions, with 50 directions at $b = 1000$ s/mm² and 50 directions at $b = 2000$ s/mm². Preprocessing imaging quality control involves automatic motion detection and expert rating of the imaging quality[25]. Multishell diffusion data that passed preprocessing imaging quality control were processed through forward-reverse gradient warping, gradient nonlinearity distortion correction, eddy current correction, and motion correction to reduce the spatial distortion and signal heterogeneities driven by scanner differences. The corrected images were then aligned to a common atlas using rigid-body registration, adjusting the diffusion gradient directions to account for head rotation relative to the atlas[25]. Fiber orientation density (FOD) functions were calculated for each voxel, and the derived tensor information together with T1 structural information was fed into multi-channel nonlinear smoothing spline registration, resulting in positional and orientational aligned voxel-wise diffusion data in 2 mm resolution. Post-processing quality measures were calculated based on the voxelwise correlations between registered images and synthesized imaging metrics given the common atlas. Images with average correlations to the atlas below 0.8 were excluded.

Restriction spectrum imaging (RSI) models the diffusion signals as mixtures of spherical harmonic basis functions[5,12]. Based on the intrinsic diffusion characteristics of separable pools of water in the human brain (i.e. intracellular, extracellular, and unhindered free water), RSI estimates the signal volume fractions of each compartment and their corresponding spherical harmonic coefficients. The measure of restricted isotropic diffusion (N0) is the coefficient of the zeroth-order spherical harmonic coefficient, normalized by the Euclidian norm of all model coefficients. This feature is most sensitive to isotropically diffusing water in the restricted compartment, within cell bodies. The measure of restricted directional diffusion (ND) is the sum of second and fourth-order spherical harmonic coefficients, normalized by the norm of all model coefficients. This feature is sensitive to anisotropically diffusing water in the restricted compartment, within oriented structures such as axons and dendrites. The normalized free water diffusion (NF) measure is calculated as the zeroth-order spherical harmonic coefficients for the unhindered water compartment. NF is also normalized by the Euclidean norm of all-spherical harmonics coefficients. This normalization makes the RSI features unitless and in the range of 0 to 1.

**Genotype data processing**. For UKB, we used the released v3 imputed genotype data. For ABCD, we used the public release 3.0 imputed genotype data. Both datasets were imputed with the HRC reference panel[74]. We performed post-imputation quality control to only allow for GWAS on common bi-allelic SNPs. We filtered SNPs which have minor allele frequencies less than 0.5 percent, Hardy-Weinberg disequilibrium (p < 1e-10), and missingness greater than 5 percent. Genetic principal components and ancestral factors were derived using well-called independent SNPs for both datasets and were used for controlling population stratification in our analyses.

**Combined principal component GWAS (CPC)**. In the present multivariate GWAS of RSI measures we implemented the CPC method[22] in the MOSTest package[21]. When the covariance among input measures is identity, CPC testing statistic is mathematically equivalent to the MOSTest test. Therefore, the codebase needed for performing CPC on ultra-high dimensional imaging data is compatible to our MOSTest except the following two components: First, the imaging measures were undergoing eigen-decompostions to derive PCs. Second, the testing statistics were based on close-form solution instead of the permutation scheme. As we were working on identity covariance matrix with finite number of PCs instead of million of voxels, CPC is a practical alternative to the original MOSTest.

CPC has been shown to be a robust multivariate GWAS method that is well powered to detect loci across different scenarios[19,22,29]. In our case, we optimized our power to detect genetic variants that shape the brain development, leaving traces in multiple brain regions. CPC enables the identification of loci that have association signals across multiple PCs, without the caveats of focusing on single brain regions. The procedures were as follows. First, the PCs and their corresponding eigenvectors were derived given the voxel-wise imaging data (Supplementary Figure 2-9). Each SNP was regressed on each of the derived PC scores, controlling for age, sex, 20 genetic PCs, genotyping batches, and intracranial volume. For a given SNP, the Wald statistics for each PC were combined as a simple linear sum (Fig. 1b). Given that PCs are orthonormal, the sum of the squared Wald statistics follows the $\chi^2$ distribution with k degrees of freedom for k PCs combined[19,22]. Although several different combination functions can be used[19], we found the global-local combination with Fisher's method proposed in the original CPC paper has greatest power in detecting genetic loci[22]. Therefore, we experimented with four different global-local cut points (50, 100, 500, and 1000 PCs) to see which combinations yield the most discoveries. To reflect this experiment, we lowered the significance threshold to $p < 4.2e-9$ (corrected for 12 multiple comparisons, as 4 thresholds and 3 features were used in the current study).

**Validation with confirmatory polyvoxel scoring**. To perform the validation test for the discovered loci, we used the confirmatory polyvoxel scoring instead of repeating the GWAS on the independent cohorts. The eigenvectors ($v_k$) and the regression coefficients ($\beta_k$) obtained from the discovery set were used to calculate the imaging scoring for all subjects in the validation sets.

$$score = \sum \beta_k v_k x' \qquad (1)$$

x stands for the raw imaging data. Given that each PC is independent of the other, it can be shown that the SNP regression on the polyvoxel score is equivalent to the comparison of the consistencies of regression coefficients between the discovery set and validation set.

**Regional enrichment for spatial distribution across voxels**. To provide more interpretability for the multivariate GWAS results, we developed a regional enrichment analysis to show which brain regions have relatively stronger signals. Most previous imaging studies relied on re-doing the voxelwise association tests to show the effect distributions of the discovered loci[14,16–18,36]. Given the distributed nature of the effect sizes among imaging measurements, the voxel-wise associations were not an ideal way of localizing effects[20]. Instead, we examined the overlap between association patterns and regions of interest in the co-registered anatomical atlas. The enrichment score is the probability-weighted regression coefficients from CPC:

$$score = \frac{\sum_i P_i \hat{\beta}_i}{\sum_i P_i} \qquad (2)$$

The variance of the enrichment score was estimated by bootstrapping the association patterns from SNPs that did not surpass the significance threshold. We then calculate the corresponding enrichment z-score and the corresponding p-values. In the current study, we obtained 130 probability maps of brain regions defined in the common atlas (Supplementary Data 9). We applied the regional enrichment analyses on the loci that showed robust signals across adult and adolescent data.

**Loci annotations, overlaps, and gene-set enrichment analyses**. To annotate the identified genetic loci, we used FUMA[39] and the GRanges function in R. SNPs with LD of r2 < 0.1 and within 250 kb distance were considered as one single locus. MAGMA[40] was used for calculating the gene-set enrichment. To map the candidate genes onto the identified loci, we used FUMA with Hi-C mapping and eQTL information from PsychENCODE[75].

**Calculation of high dimensional heritability**. Previous studies on the heritability of high-dimensional phenotypes indicated the average heritability is a valid way of estimating the genetic architecture of human traits[23,76]. It is equivalent to the weighted average of heritabilities across each of the PCs. We applied LD score regression for each PC and then weighted these according to their eigenvalues, deriving the average heritabilities across RSI features.

**Stratified LD score regression for heritability enrichment analyses**. As the prior literature on multivariate GWAS has demonstrated[23], the multivariate $\chi^2$ can be rescaled and then used with stratified LDSC (S-LDSC) to examine the relative enrichment of heritability for given annotations. Here, we examined the tissue-specific enrichment through histone marker annotations of human tissues, given that the regulatory landscape has more tissue specificity than gene expressions[57]. For cell-type specific analyses, we used the mouse single cell ATAC-seq data because it is a comprehensive resource with established utility in prioritizing human risk variants[58]. The scaled genome-wide multivariate $\chi^2$ for each imaging metric, i.e. N0, ND, and NF, was regressed against the tissue-specific/cell-type-specific annotations, while controlling for the baseline annotations as recommended by S-LDSC[57]. We reported the signed enrichment Z statistics, as well as the corresponding multiple comparisons adjusted p values.

**Calculation of shared genome-wide signals between two phenotypes**. As proposed in other multivariate GWAS efforts[23], for a given summary statistics of a phenotype, we first calculated the average magnitudes of associations in each of the approximately independent LD blocks[77], deriving the unsigned polygenic signal profiles of a given trait. Spearman correlations were performed for each pair of the GWAS results, evaluating the level of overlapping in the genome-wide signals.

**Reporting summary**. Further information on research design is available in the Nature Research Reporting Summary linked to this article.

## Data availability

Data from UKB is available through UKB application (https://www.ukbiobank.ac.uk). The research has been conducted using the UK Biobank Resource under Application Number 27412. Adolescent data used in the preparation of this article were obtained from the Adolescent Brain Cognitive Development℠ Study (ABCD Study®) (https://abcdstudy.org), held in the NIMH Data Archive (NDA). ABCD data used in here is under the NDA study registered at https://doi.org/10.15154/1524729. Genomic locus and gene-set results can be found in the Supplementary Data. Full summary statistics can be found in LocusZoom.js[78] (N0: https://my.locuszoom.org/gwas/575925/; ND: https://my.locuszoom.org/gwas/611203/; NF: https://my.locuszoom.org/gwas/644492/).

## Code availability

ABCD processing codes can be found in github repository series (https://github.com/ABCD-STUDY). Codes used specifically for this study, including obtaining restricted spectrum imaging metrics, combined principal components GWAS, polyvoxel scores, and spatial regional enrichment analyses, can be found in the public accessible GITHUB page at (https://github.com/cmig-research-group/RSIGWAS). The code version used in this study is registered[79]. The main code base is on MATLAB version 2017b.

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

## Acknowledgements

This work was supported by grant R01MH122688, RF1MH120025, and R01MH118281 funded by the National Institute for Mental Health (NIMH). Data used in the preparation of this article were obtained from the Adolescent Brain Cognitive Development ^SM(ABCD) Study (https://abcdstudy.org), held in the NIMH Data Archive (NDA). The ABCD Study® is supported by the National Institutes of Health and additional federal partners under award numbers U01DA041048, U01DA050989, U01DA051016, U01DA041022, U01DA051018, U01DA051037, U01DA050987, U01DA041174, U01DA041106, U01DA041117, U01DA041028, U01DA041134, U01DA050988, U01DA051039, U01DA041156, U01DA041025, U01DA041120, U01DA051038, U01DA041148, U01DA041093, U01DA041089, U24DA041123, U24DA041147. A full list of supporters is available at https://abcdstudy.org/federal-partners.html. A listing of participating sites and a complete listing of the study investigators can be found at https://abcdstudy.org/consortium_members/. ABCD consortium investigators designed and implemented the study and/or provided data but did not necessarily participate in the analysis or writing of this report. This manuscript reflects the views of the authors and may not reflect the opinions or views of the NIH or ABCD consortium investigators. The ABCD data repository grows and changes over time. The ABCD data used in this report came from https://doi.org/10.15154/1524729. The fast track data release used in this report are available at https://nda.nih.gov/edit_collection.html?id=2573. Instructions on how to create an NDA study are available at https://nda.nih.gov/training/modules/study.html). We specially thank Megan Chang for her assistance on organizing and curating the genomic relevant documents.

## Author contributions

C.C.F., A.M.D., and O.A.A. conceptualized and designed the study. C.C.F. and R.L. performed the analyses. D.J.H. and O.F. processed the data. C.C.F. interpreted the results and wrote the draft of the manuscript. R.L., C.M., D.P., C.H.C., D.J.H., W.K.T., N.P., D.M., O.F., O.A.A., and A.M.D. provide critical inputs for the revision of manuscripts.

## Competing interests

Dr. Andreassen has received speaker's honorarium from Lundbeck and Sunovion, and is a consultant to HealthLytix. Dr. Dale is a Founder of and holds equity in CorTechs Labs, Inc, and serves on its Scientific Advisory Board. He is a member of the Scientific Advisory Board of Human Longevity, Inc. and receives funding through research agreements with General Electric Healthcare and Medtronic, Inc. The terms of these arrangements have been reviewed and approved by UCSD in accordance with its conflict of interest policies. The other authors declare no competing interests.
