## [Peer Review File · Nature Communications]

Multivariate genome-wide association study on tissue-sensitive diffusion metrics highlights pathways that shape the human brainREVIEWER COMMENTS

Reviewer #1 (Remarks to the Author):

This manuscript describes a large-scale imaging genomic study using GWAS to identify genes related to variation in three in vivo measures of brain tissue composition from multi-shell diffusion imaging: N0, ND and NF. For each phenotype, the authors implement a combined principal components GWAS (CPCs) that uses genetic associations with PCs of the person*voxel data for that phenotype to generate a summary association score per SNP. The authors conduct their primary analyses in the UKB sample, and report overlap of associated loci between N0, ND and NF, as well as conducting GO enrichments for associated genes. Replicability of findings is assessed using a “polyvoxel score” method on a more recent set of UKB scans and scans from the ABCD study. Validated loci are then taken forwards for comparison with significant loci from past imaging-genetics analyses. Regional effects are examined based on a “regional enrichment analysis” with focused discussion of a handful of loci that the authors state showed consistent regional enrichment across datasets. The manuscript ends with an analysis of overlap between associated loci and databases of “druggable targets” and regional/cell-type genomic annotations.

The questions addressed in this study are well-motivated. Measures from multi-shell diffusion data provide a unique insight into tissue composition that is lacking from more mainstream morphometric indices from in vivo imaging, and there is value in better understanding the genetic determinants of variation in these phenotypes. Other strengths of the manuscript include use of a large UKB primary sample, robust correction for multiple comparisons across CPC GWAS parameter search on top of across the genome, efforts to test for replicability in independent datasets, and provision of annotation/enrichment analyses. Overall, I think this work is potentially of interest and value to the field, but some treatment of the issues below would be important in fully realizing this potential.

I am not sufficiently familiar with the CPC GWAS method, the polyvoxel score approach to testing for replicability, and the regional enrichment tests, or the mathematical principles upon which these tests depend, to provide an expert commentary on their internal validity. The authors are leaders in methods-development for statistical genetics (especially with highly multivariate phenotypes such as these), so I think the prior probability of these methods being sound is high. However the complexity of each method, and the chaining of multiple complex methods to support the inferences being made is such that expert statistical review of this manuscript would - in my opinion - be worthwhile.

In reviewing this manuscript, I thought the following issues might benefit from further consideration:

Linking to “druggable targets”:

Very little information is provided regarding the database used for this analysis, and the definition(s) of “druggable” upon which it is based. Given that the method for finding genetic associations is highly abstracted and information re e.g. the directions and regions of effects can only be queried on a selective basis, I found it very hard to understand what exactly one could infer in concrete terms from an alignment with the DGIdb. The authors own data show that influences on the DWI metrics are highly polygenic, and these complex combined small effects are operating over developmental time. Amidst this complexity, it is not clear to me what it means to know that any single locus can in isolation be influenced by a drug. There is so much to qualify and unpack before the actionable meaning (if any) of such a finding can be understood. For this reason I would suggest changing the last sentence of introduction to something more like “Our results shed new light on the biological determinants of brain tissue compositional differences in health, and potential overlaps between these and the genetic basis of neuropsychiatric disease”. I would also remove the DGIdb entirely, or place it in supplementary material, or substantially expand the manuscript to more fully consider the meanings and caveats of such an analysis. My concern is that statements like “The Drug-Gene Interaction database (DGIdb) shows VCAN as tier one druggable target and was found to be interacting with cyclosporine indicating a potential path for pharmacological interventions” are leaping too far ahead of a database overlap.

Approach to defining replication set of UKB scans:

The authors used a 2019 cut off date to distinguish the 25k primary UKB set from the 6k validation set, stating that “We decided to use this naturally occurring cut-point instead of randomized allotment of the groups because of best practice considerations avoiding potential systematic biases driven by temporally related imaging confounds. “. I may be musing something (in which case perhaps some clarification in text would also help other readers), but would this temporal cut off not increase the risk of temporal biases?

Image QA and QC:

No information is given on this and details would be important to provide.

Understanding the imaging-based PCs that are used to tests for genetic associations:

Sup Fig 2 was notable for showing that the first ~5 PCs to have lower h^2 than subsequent PCs. Heritability seemed to peak at ~PC10. Is this mathematically expected from the method (not sure, but I think not), and if not – would this observation not warrant further investigation? More generally, is there not a need to be reasonably confident that none of the PCs included capture e.g. motion-related noise? I appreciate that locus-level associations are reflective of the combined influence across PCs, but this property of the method would not protect it from finding genetic associations with e.g. motion-related imaging features. For example, do any of the PCs capture (presumably meaningless) phenotype variation in the ventricles? The 50 PC version is amenable to visual inspection and I think some comment on this would be important. Also, does the method allow for detection of spatial PCs that weigh heavily in driving the summary association? If so, it might be good to identify and visualize these (in paper).

Regional enrichment analyses:

Sup Table 10 seemed to just be listing the ROIs rather than providing regional enrichment test statistics. Could regional statistics be reported, and perhaps correlated across regions between DWI phenotypes?

Enrichment analysis against tissue-specific chromatin annotations and cell type-specific annotations:

These sections highlight a general issue in the manuscript, of presenting annotation analysis at a very high-level, with limited information being provided in the main methods or supplemental materials, and little interpretation of what the findings can and can't say. I appreciate that space is limited given the questions being tackled, but the net effect is a flurry of associations with unclear solidity and meaning. For example, why were tissue histone markers vs. eQTLs used? Why were mouse ATACseq data used for cell-type annotation rather than annotations (not necessary ATAC-seq based) from human single nucleus/cell studies? I'm not raising these points to suggest that the wrong choices were made, but rather than there needs to be some discussion for why analyses were conducted in the specific way they were. Greater detail/transparency would be good in the methods description too (for example the murine basis for cell markers was only apparent when I read the cited paper).

Expanded Discussion:

This would be good to provide some interpretation and discussion of caveats/limitations for findings which seemed to be a bit lacking overall.

Reviewer #2 (Remarks to the Author):

This manuscript uses multivariate GWAS to identify variants associated with principal components of different tissue composition estimates obtained from diffusion MRI. It relies on large, publicly available datasets and a strength is the replication of some key findings across Biobank and ABCD. I am no expert on GWAS methodology but the general approach seems sound. The primary innovation is the multivariate analysis. Various informatics approaches are then used to generate some general insights about the disease lists.

The phenotypes considered rely on a model-based estimates of neural tissue composition. To what extent to the models accurately describe the data? The authors should provide some markers of model fit/accuracy/validity so that readers can judge their adequacy.

Lines 124 – 128 are difficult to unpack. Please define multi-dimensional heritability. It is also unclear how the authors conclude that ~60% of the average NSP heritabilities are explained by the discovered loci.

Line 156 – what is the evidence for a spatial gradient and what does it look like?

Lines 201 – 213 – please explain what it means for a locus to be “druggable” and how this is determined.

Lines 235 – 253 – the associations with disorder results are weak and conclusions should be tempered accordingly.

Line 251 – Pleiotropy is not demonstrated here. It is possible that the imaging measures mediate the link between genes and disease.

Line 259 – this study does not really demonstrate the specific impact of molecular pathways – it simply interprets associations. No interventional studies are performed.

Reviewer #3 (Remarks to the Author):

Manuscript Review

Multivariate genome-wide association study on tissue-sensitive diffusion metrics identifies key molecular pathways for axonal growth, synaptogenesis, and astrocyte-mediated neuroinflammation
Fan et al., 2021

Overview

Authors carried out GWAS for three novel neuroimaging phenotypes N0, ND and NF. N0 is most sensitive to anisotropically diffusing water in within cell bodies. ND is most sensitive to anisotropically diffusing water within oriented structures such as axons and dendrites. NF is a proxy measure of free water component. Authors carried out validation using polygenic voxel scores and carried out a series of downstream analysis to characterize the genomic loci discovered. Authors report that they identified key molecular pathways involved in axonal growth, astrocyte-mediated neuroinflammation, and synaptogenesis during development. Additionally, they reported drug annotations of the primary GWAS findings for potential targets for pharmacological intervention

Overall impression

The paper addresses an important issue in neuroimaging genomics looking at diffusion measures. The phenotypes that the authors reported on would potentially advance the field in terms of how we understand connectivity in the brain. In addition, authors have used an interesting method that address pleiotropy across large number of neuroimaging measures to distill a composite GWAS of each connectivity-based neuroimaging phenotypes. Nonetheless, there are several queries that I have with regards to the approach to the results of the association analysis. In part, there is probably not enough details provided to the reader to fully appreciate the findings that the authors are reporting on. A minor point – if authors could format some of the Supplementary Tables in a spreadsheet format it would be very much more helpful to query some of the results that they are reporting on.

Specific Queries

Phenotypic Distributions and Descriptive Statistics

One of the novel aspects of the current report (and its strength) is in the neuroimaging phenotypes investigated. Would the authors be able to provide codes/scripts to demonstrate exactly how the phenotypes were derived from DTI measures? Related to that question, would authors also comment on the distribution of the derived phenotypes and how they related to the usual FA measures that are usually reported? Preferably, the results would have been more complete if the authors reported some descriptive statistics regarding their novel phenotypes N0, ND, NF and provided some visualization for the data.

Data Analysis

Authors have indicated that the “multi-shell” method would be a more sensitive approach in understanding the biological underpinnings of diffusion neuroimaging measures. Authors report that the data analysis started with 100 x 100 x 130 voxels. It was then reported that the data was reduced to 5000PCs. I’m not quite sure how to reconcile the numbers reported in the methods where 50, 100, 500, and 1000 principal components were experimented with – I might have missed the conceptual decision to select 5000PCs.

I guess the question comes up why 5000PCs - why not more or why not less? Were there simulations carried out to suggest that 5000PCs is the optimal number? Authors cited MOSTest as the “available code” but I’m not so sure if MOSTest was utilized in this context? That brings up another question, why did the authors decided to go with the CPC method as opposed to other methods?

For the CPC approach, each SNP was regressed (with covariates) against each neuroimaging PC as phenotype. Which suggest for N0, ND, NF, 5000 GWASs were carried out, and the effect sizes were combined using the Wald statistics and df to generate a p-value for the association. The challenge with this type of pleiotropic methodology is that Wald statistics is not likely to have association direction. Just the magnitude. It would also mean that the method would not yield a heterogeneity measure, but in a way “collapse” both heterogeneity p-values and association p-values to give a much more powered form of association. If that’s the case, it would be challenging to interpret the results of the “genetic overlap” even for other traits, or even fully show how each gene might be a potential “drug target” without the effect size for the variant association.

Results

Loci Discovery:

Authors report that a total of 503 unique loci across three of the neuroimaging measures in the report. Of these, 152 had not been previously reported. I don’t see a table where these loci were specifically reported or annotated. These are probably in separate supplementary tables that the authors indicated (Supp Tables 2-7). It is recommended that authors present these novel loci in a separate table that are annotated so that readers could have a chance to evaluate the results of the findings. In addition, it would be good to have the source data for the non-novel loci reported in Figure 2d. Again, this would give the reader a chance to evaluate the findings presented in the report.

Validation:

Authors used a ‘polyvoxel approach’ to validate the findings of the discovery findings. This would involve summing across voxels weighted by the regression weight and eigenvector. I assume that coefficient k represents each voxel. How many voxels were included in this analysis? Relatedly, would authors see an association in their independent datasets without weighting on the regression coefficient – just using the eigenvector weights? And would the association effects/significance change after weighting – would the models be significant if compared against null > eigenvector weighted only > eigenvector + regression weights from discovery set.

Authors also attempted to examine enrichment in brain regions using the polyvoxel approach. The regions were included in Supplementary Table 11 (not Supplementary 10 as indicated in the manuscript). There was no further discussion about this analysis in the results.

Code/Data Availability

Authors cited MosTest and ABCD pipelines as available code. However, it would be critical for some of the computational pipelines for the polyvoxel approach and estimation of N0, ND, and NF to be available as well since these are the aspects that make the report novel.

RESPONSE TO REVIEWERS

Reviewer #1 (Remarks to the Author):

This manuscript describes a large-scale imaging genomic study using GWAS to identify genes related to variation in three in vivo measures of brain tissue composition from multi-shell diffusion imaging: N0, ND and NF. For each phenotype, the authors implement a combined principal components GWAS (CPCs) that uses genetic associations with PCs of the person*voxel data for that phenotype to generate a summary association score per SNP. The authors conduct their primary analyses in the UKB sample, and report overlap of associated loci between N0, ND and NF, as well as conducting GO enrichments for associated genes. Replicability of findings is assessed using a “polyvoxel score” method on a more recent set of UKB scans and scans from the ABCD study. Validated loci are then taken forwards for comparison with significant loci from past imaging-genetics analyses. Regional effects are examined based on a “regional enrichment analysis” with focused discussion of a handful of loci that the authors state showed consistent regional enrichment across datasets. The manuscript ends with an analysis of overlap between associated loci and databases of “druggable targets” and regional/cell-type genomic annotations. The questions addressed in this study are well-motivated. Measures from multi-shell diffusion data provide a unique insight into tissue composition that is lacking from more mainstream morphometric indices from in vivo imaging, and there is value in better understanding the genetic determinants of variation in these phenotypes. Other strengths of the manuscript include use of a large UKB primary sample, robust correction for multiple comparisons across CPC GWAS parameter search on top of across the genome, efforts to test for replicability in independent datasets, and provision of annotation/enrichment analyses. Overall, I think this work is potentially of interest and value to the field, but some treatment of the issues below would be important in fully realizing this potential. I am not sufficiently familiar with the CPC GWAS method, the polyvoxel score approach to testing for replicability, and the regional enrichment tests, or the mathematical principles upon which these tests depend, to provide an expert commentary on their internal validity. The authors are leaders in methods-development for statistical genetics (especially with highly multivariate phenotypes such as these), so I think the prior probability of these methods being sound is high. However the complexity of each method, and the chaining of multiple complex methods to support the inferences being made is such that expert statistical review of this manuscript would - in my opinion - be worthwhile.

RESPONSE:

We sincerely thank the reviewer for the encouraging comments and suggestions. We revised our manuscript accordingly to enhance the transparency and clarity of our approach and results, which we documented in the following responses.

In reviewing this manuscript, I thought the following issues might benefit from further consideration:

- Linking to “druggable targets”:

Very little information is provided regarding the database used for this analysis, and the the definition(s) of “druggable” upon which it is based. Given that the method for finding genetic associations is highly abstracted and information re e.g. the directions and regions of effects can only be queried on a selective basis, I found it very hard to understand what exactly one could infer in concrete terms from an alignment with the DGIdb. The authors own data show that influences on the DWI metrics are highly

polygenic, and these complex combined small effects are operating over developmental time. Amidst this complexity, it is not clear to me what it means to know that any single locus can in isolation be influenced by a drug. There is so much to qualify and unpack before the actionable meaning (if any) of such a finding can be understood. For this reason I would suggest changing the last sentence of introduction to something more like “Our results shed new light on the biological determinants of brain tissue compositional differences in health, and potential overlaps between these and the genetic basis of neuropsychiatric disease”. I would also remove the DGIdb entirely, or place it in supplementary material, or substantially expand the manuscript to more fully consider the meanings and caveats of such an analysis. My concern is that statements like “The Drug-Gene Interaction database (DGIdb) shows VCAN as tier one druggable target and was found to be interacting with cyclosporine indicating a potential path for pharmacological interventions” are leaping too far ahead of a database overlap.

RESPONSE:

We thank the reviewer for the insightful comments and suggestions. Indeed, we are still far from translating GWAS discoveries to actual clinical practice, although we can start to lay out meaningful biological hypotheses based on evidence gleaned from GWAS. However, as the reviewer suggested, our discussion on potential druggable targets may be hard to understand or misinterpreted by readers. Therefore, we revised our manuscript to reduce the emphasis on potential pathways for interventions. Following the reviewer’s comment, we moved the entire section on the druggable targets to the Supplemental Information and revised the following sentences throughout the manuscript to address the issue of potential over-interpretation:

In the last two sentence of the Abstract:

"Key molecular pathways involved in axonal growth, astrocyte-mediated neuroinflammation, and synaptogenesis during development were found to significantly impact the measured variations in tissue-specific imaging features. Our results shed new light on the biological determinants of brain tissue composition and their potential overlap with the genetic basis of neuropsychiatric disorders."

In the last two sentences of the Introduction:

"By investigating the spatial distribution of the associated effects, we highlighted critical molecular pathways involved in neuroinflammation and axonal growth, and the corresponding regions that may be susceptible to these processes. Signal overlap, at both the locus level and genome-wide, with neuropsychiatric outcomes indicate the functional relevance of our GWAS results, providing a foundation for further understanding of the biological underpinnings of neuropsychiatric disorders."

In the discussion of VCAN in the results section:

"VCAN, which encodes versican and is a lectican-binding chondroitin sulfate proteoglycan (CSPG), serves a critical role in astrocyte-mediated neuroinflammation⁴¹, and has potential interacting pharmacological targets^{42,43} (Figure 3d; Supplemental Information). ... Changes in the distribution of CSPGs in the hippocampal formation were observed among patients with schizophrenia and patients with bipolar disorder^{45,46}, linking our findings to neuropsychiatric outcomes."

In the discussion of DPYSL5 in the results section:

Our findings are also relevant to neuropsychiatric outcomes, as CRMP has been implicated in schizophrenia and mood disorders⁵⁰.

- Approach to defining replication set of UKB scans:

The authors used a 2019 cut off date to distinguish the 25k primary UKB set from the 6k validation set, stating that “We decided to use this naturally occurring cut-point instead of randomized allotment of the groups because of best practice considerations avoiding potential systematic biases driven by temporally related imaging confounds. “. I may be musing something (in which case perhaps some clarification in text would also help other readers), but would this temporal cut off not increase the risk of temporal biases?

REPOSE:

We thank the reviewer for giving us a chance to clarify the rationale for selecting the replication set in UKB. As a recent paper by Bradley Efron pointed out (Efron, 2020, Journal of the American Statistical Association), when it comes to the training/testing data split for examining the generalizability of models, random allocation might lead to over-estimation of the generalizability whereas the time based data split might be closer to a realistic scenario. It means that if there was any drift of the study design, such as imaging software updates, the signals related to that drift would not generalize in the time-split samples. Given our focus on finding the genetic signals relevant to brain tissue composition, we erred on the conservative side to make sure the replication tests are robust in terms of their generalizability. In the revised manuscript, we elaborated on the reasons behind our decisions in the results and method section as the following:

In the result section, first paragraph:

"To ensure the validating test was robust against study variability due to time shift²⁸, we selected UKB samples received MRI scans before 2019 as the discovery set while all others were regarded as replication sets."

In the first paragraph of the method section:

"We decided to use this naturally occurring temporal cut-point instead of randomized allotment of the groups because of best practice considerations^{20,28,68,69}, avoiding potential systematic biases driven by temporally related imaging confounds. In particular, the potential time shift of the study design can lead to over-optimistic evaluation on the generalizability if random data split instead of time split was used²⁸. Given our purpose to discover and validate biologically relevant effects, we used a conservative approach by selecting a naturally occurring time point as the selection criteria for discovery and replication sets in UKB."

- Image QA and QC:

No information is given on this and details would be important to provide.

RESPONSE:

We added the following descriptions on image quality assessment and quality control in the method section:

" To harmonize the imaging data across the two studies, we processed the dMRI data from UKB and ABCD using the ABCD-consistent imaging processing pipeline implemented by the ABCD Data Analysis, Informatics, and Resource Center (ABCD DAIRC). The detailed processing procedures have been published elsewhere ²⁵. In short, multi-shell diffusion MRI data of ABCD acquired with seven $b=0$ s/mm² frames and 96 non-collinear gradient directions, with 6 directions at $b=500$ s/mm², 15 directions at $b=1000$ s/mm², 15 directions at $b=2000$ s/mm², and 60 directions at $b=3000$ s/mm². Multi-shell diffusion MRI data of UKB acquired with five $b=0$ s/mm² frames and 100 non-collinear gradient directions, with 50 directions at $b=1000$ s/mm² and 50 directions at $b=2000$ s/mm². Preprocessing imaging quality control involves automatic motion detection and expert rating of the imaging quality ²⁵. Multi-shell diffusion data that passed preprocessing imaging quality control were processed through forward-reverse gradient warping, gradient nonlinearity distortion correction, eddy current correction, and motion correction to reduce the spatial distortion and signal heterogeneities driven by scanner differences. The corrected images were then aligned to a common atlas using rigid body registration, adjusting the diffusion gradient directions to account for head rotation relative to the atlas ²⁵. Fiber orientation density (FOD) functions were calculated for each voxel, and the derived tensor information together with T1 structural information was fed into multi-channel nonlinear smoothing spline registration, resulting in positional and orientational aligned voxel-wise diffusion data in 2 mm resolution. Post-processing quality measures were calculated based on the voxel-wise correlations between registered images and synthesized imaging metrics given the common atlas. Images with average correlations to the atlas below 0.8 were excluded. "

- Understanding the imaging-based PCs that are used to tests for genetic associations:

Sup Fig 2 was notable for showing that the first ~5 PCs to have lower h^2 than subsequent PCs. Heritability seemed to peak at ~PC10. Is this mathematically expected from the method (not sure, but I think not), and if not – would this observation not warrant further investigation? More generally, is there not a need to be reasonably confident that none of the PCs included capture e.g. motion-related noise? I appreciate that locus-level associations are reflective of the combined influence across PCs, but this property of the method would not protect it from finding genetic associations with e.g. motion-related imaging features. For example, do any of the PCs capture (presumably meaningless) phenotype variation in the ventricles? The 50 PC version is amenable to visual inspection and I think some comment on this would be important. Also, does the method allow for detection of spatial PCs that weigh heavily in driving the summary association? If so, it might be good to identify and visualize these (in paper).

RESPONSE:

We thank the reviewer for giving us the opportunity to elaborate on the statistical intricacies of the genetic associations with phenotypic principal components (PCs). Yes, unlike the monotonic decrease of the eigenvalues (Supplemental Figure 6), the heritability estimates on the PCs were not proportionally scaled (Supplemental Figure 11). For example, N0 has the highest heritability on first PC whereas ND and N0 have the highest heritabilities in later PCs. It is because eigenvalue decompositions were performed on the phenotype measures without the knowledge of genetic information; thus, the resulting eigenvalues were orthogonal basis functions in the "phenotypic space", not in the "genetic space". It means the resulting PC scores were blind to the underlying genetic contributions. Only after the associations with genetic variants were performed, the obtained regression coefficients/weights provided an idea of how much of the genetic information each PC contains. A similar phenomenon has been

observed in the applications of PC-based GWAS (See reference 23, Extended Data Fig. 8, heritability estimates of principal components of face data).

Furthermore, each PC should be regarded as a basis function, where no single PC has a direct interpretation. To illustrate this point, we rendered the mid coronal section of the first 64 PCs of N0, ND, and NF (Supplemental Figure 7-9). The eigenvectors were not necessarily confined to certain anatomical structures. Similar to discrete Fourier transformation, any complex spatial distribution can be represented as the sum of series of simple spatial bases. Similarly, each PC only described a fraction of the phenotypic variations and the spatial distribution of a given genetic effect would only be revealed by the weighted combinations of eigenvectors. For example, one of the genetic variants we highlighted, rs12653308, has much stronger bilateral effects in the hippocampus, yet no single PC has evident signals confined to such anatomical structures (Figure 3b-c; Supplemental Figure 7). The combined PC approach is designed for identifying genetic loci that have evident signals across PCs and help prevent the detection of nuisance effects.

Nevertheless, despite the statistical advantages of combined PC GWAS, we exercised caution when performing our analyses. First, the images have been corrected for motion, eddy current distortions, and intensity inhomogeneity to reduce the impact of systematic noise. Second, we included intracranial volume as a covariate in regression analyses. Third, we masked the imaging regions that contain no brain tissue, such as the ventricles. Finally, we did not impose any sparsity constraint on how the PC is combined, avoiding weighting too much on a small set of PCs and complicating the multiple comparisons. As our results show, the identification of biologically relevant variants and the signal enrichment of the specific brain tissues provide strong support for the validity of our conservative approach.

We added the following paragraphs to elaborate the pros and cons of a combined PC approach:

In the result section:

" Using the UKB discovery set, we calculated the principal components (PCs) from the tissue feature across all voxels"

" Each of the PCs can be regarded as an orthogonal basis function with limited interpretability, yet the weighted combination of them can represent any spatial distribution (Supplementary Figure 7-9). CPC combined the association signals across PC for a given genetic variant and detect the genetic loci that are shared across multiple PCs, thus reducing the burden of multiple testing and the false detection of nuisance effects."

In the discussion section:

" Because our multivariate GWAS was optimized on detecting signals shared across PCs, the statistical power may be less than ideal for detecting extremely sparse genetic effects, i.e. limited to only one or two PCs. Although it is possible to have regionally specific genetic effects, it is less likely in our case since our PCs captured information across the whole brain and were anatomically agnostic. Instead of having one PC to represent one particular anatomical structure, it was the weighted combinations of several PCs that highlighted certain anatomical structures. This is the benefit of using CPC, as it implicitly picks up the patterning signals without pre-defining the region of interest. However, these statistical properties can also make it difficult to interpret which anatomical regions were most relevant"

for discovered loci. To facilitate interpretation, we implemented regional enrichment analyses, examining which anatomical structures have higher average signals compared to other regions. "

In the method section:

" CPC has been shown to be a robust multivariate GWAS method that is well-powered to detect loci across different scenarios ^{19,22,29}. In our case, we intended to optimize our power to detect genetic variants that shape brain development, leaving traces in multiple brain regions. CPC enables the identification of loci that have association signals across multiple PCs, without the caveat of focusing on single brain regions. The procedures were as follows. First, the PCs and their corresponding eigenvectors were derived given the voxel-wise imaging data (Supplemental Figure 3-5). Each SNP was regressed on each of the derived PC scores, controlling for age, sex, 20 genetic PCs, genotyping batches, and intracranial volume."

- Regional enrichment analyses:

Sup Table 10 seemed to just be listing the ROIs rather than providing regional enrichment test statistics. Could regional statistics be reported, and perhaps correlated across regions between DWI phenotypes?

RESPONSE:

We added the regional enrichment testing statistics for each locus in the supplemental tables.

- Enrichment analysis against tissue-specific chromatin annotations and cell type-specific annotations:

These sections highlight a general issue in the manuscript, of presenting annotation analysis at a very high-level, with limited information being provided in the main methods or supplemental materials, and little interpretation of what the findings can and can't say. I appreciate that space is limited given the questions being tackled, but the net effect is a flurry of associations with unclear solidity and meaning. For example, why were tissue histone markers vs. eQTLs used? Why were mouse ATACseq data used for cell-type annotation rather than annotations (not necessary ATAC-seq based) from human single nucleus/cell studies? I'm not raising these points to suggest that the wrong choices were made, but rather than there needs to be some discussion for why analyses were conducted in the specific way they were. Greater detail/transparency would be good in the methods description too (for example the murine basis for cell markers was only apparent when I read the cited paper).

RESPONSE:

We thank the reviewer for the comments and suggestions on the clarity of our enrichment analyses. To provide more technical details, we added the following description in the methods section:

"Stratified LD score regression for heritability enrichment analyses

As prior literature on multivariate GWAS has demonstrated ²³, the multivariate χ^2 can be rescaled and then used with stratified LDSC (S-LDSC) to examine the relative enrichment of heritability for given annotations. Here, we examined tissue specific enrichment through histone marker annotations of human tissues, given that the regulatory landscape has more tissue specificity than gene expression ⁵⁷. For cell-type specific analyses, we used the mouse single cell ATAC-seq data because it is a comprehensive resource with established utility in prioritizing human risk variants ⁵⁸. The scaled

genome-wide multivariate χ^2 for each imaging metric, i.e. NO, ND, and NF, was regressed against the tissue-specific/cell-type specific annotations, while controlling for baseline annotations, as recommended by S-LDSC⁵⁷. We reported the signed enrichment Z statistics, as well as the corresponding multiple comparisons adjusted p values."

- Expanded Discussion:

This would be good to provide some interpretation and discussion of caveats/limitations for findings which seemed to be a bit lacking overall.

RESPONSE:

We revised the discussion section to add the following interpretation and discussion in our manuscript:

" Our results indicate widespread pleiotropies between the development of cortical surfaces and cerebral white matter. As the patterning of the mature brain is the end results of multiple molecular processes working from differentiation of neuroprogenitor cells, migration of neurons, to synaptic prunings⁶⁸, the relevant genes are unlikely to confine their effects on one single anatomically defined region. This is in line with findings from malformations of cortical development that the germline mutations of genes involved in cell migration would lead to global malformations instead of localized lesions⁶⁹. Many of the loci we discovered and replicated were also found to be associated with other imaging modalities across cortical and subcortical regions. We are not claiming that focal effects do not exist, indeed the variants we highlighted do have locally enriched signals. Instead we suggest that it is more likely that regional specification of the human brain is the ultimate result of a complex coordination of multiple distributed molecular processes⁷⁰ rather than that single genetic variants affect single anatomical regions. For instance, the group of genes belonging to CSPG had consistent associations with NO metrics. The association signals were enriched in multiple brain regions beyond previously reported ROIs^{18,21}, especially bilateral hippocampus. This astrocyte-dependent molecular process may have more direct effects on the synaptic pruning in the hippocampal regions and then cascading down-stream to the associated fiber tracts.

Our findings showcase the need for novel analytic approaches in brain imaging genetics. Multivariate GWAS on whole brain phenotypes circumvents the potential "spotlight bias" that region-of-interest approaches are susceptible to⁷¹. Diffuse effects across brain regions and neurobiological pathways are more easily detected with this approach, as the inference is based on the total sum of the effects. Moving beyond the metrics of structural volumes or fiber orientation enabled us to detect molecular effects on brain tissue properties, identifying relevant biological pathways important for human brain development and neuropsychiatric outcomes.

Because our multivariate GWAS was optimized for detecting signals shared across PCs, the statistical power may be less than ideal for detecting extremely sparse genetic effects, i.e. limited to only one or two PCs^{19,21,22}. Although it is possible to have regionally specific genetic effects, our approach will be less sensitive to detect such effects since our PCs captured information across the whole brain and were anatomically agnostic. Instead of having one PC to represent one particular anatomical structure, it was the weighted combinations of several PCs that highlighted certain anatomical structures. This is the benefit of using a multivariate GWAS, as it implicitly picks up the patterning signals without pre-defining the region of interest. However, these statistical properties can also make it difficult to interpret which anatomical regions are most relevant for a given discovered loci. To

facilitate the interpretation, we implemented regional enrichment analyses, examining which anatomical structures have higher average signals compared to other regions.

Our results highlight the pleiotropic nature of genes involved in synaptic pruning, neuroinflammation, and axonal growth. The microglia related molecular processes were implicated in multiple brain regions across cortical and subcortical structures. The significant loci overlaps between tissue sensitive imaging metrics and psychiatric disorders implicates the etiological mechanisms beyond the neuronal growth, such as microglia mediated synaptic pruning. Our identified genes may aid in experimental studies investigating interventions for neuropsychiatric outcomes."

Reviewer #2 (Remarks to the Author):

1. This manuscript uses multivariate GWAS to identify variants associated with principal components of different tissue composition estimates obtained from diffusion MRI. It relies on large, publicly available datasets and a strength is the replication of some key findings across Biobank and ABCD. I am no expert on GWAS methodology but the general approach seems sound. The primary innovation is the multivariate analysis. Various informatics approaches are then used to generate some general insights about the disease lists.

RESPONSE:

We sincerely thank the reviewer's positive feedback on our approach and results.

2. The phenotypes considered rely on a model-based estimates of neural tissue composition. To what extent to the models accurately describe the data? The authors should provide some markers of model fit/accuracy/validity so that readers can judge their adequacy.

RESPONSE:

Indeed, the imaging metrics we used were model-based estimates of tissue composition given the characteristics of the diffusion signals. The chosen metrics, i.e. N_0 , ND , and NF , are more sensitive to tissue composition than traditional MRI metrics. They have been shown to be highly correlated with histology of brain tissues and highly informative for brain development^{3,10,11,27}. Our findings on relevant molecular pathways and cell type-specific enrichment signals further provide biological validity of such a modeling approach. On the other hand, to enhance the accuracy of our results, we applied series of imaging quality control both pre- and post- image processing. To address the issue raised by the reviewer, we revised the following paragraphs.

In the results section:

" N_0 , ND , and NF provide greater tissue specificity than the widely-used diffusion tensor metrics and have been useful for understanding the variations of cellular organization within the human brain and highly informative for human brain development^{3,10,11,27}."

In the method section:

" Fiber orientation density (FOD) functions were calculated for each voxel, and the derived tensor information together with T1 structural information was fed into multi-channel nonlinear smoothing spline registration, resulting in positional and orientational aligned voxel-wise diffusion data in 2 mm resolution. Post-processing quality measures were calculated based on the voxel-wise correlations

between registered images and synthesized imaging metrics given the common atlas. Images with average correlations to the atlas below 0.8 were excluded."

3. Lines 124 – 128 are difficult to unpack. Please define multi-dimensional heritability. It is also unclear how the authors conclude that ~60% of the average NSP heritabilities are explained by the discovered loci.

RESPONSE:

We thank the reviewer for the comment on the heritability estimates. Although several papers have discussed the definition and the optimal solution to the multi-dimensional heritability, it can be counter intuitive to traditionally defined heritability. Therefore, we elaborated the term multi-dimensional heritability and moved this section to the Supplemental Information.

" Average heritability across three RSI metrics

Although each phenotypic PC can have heritabilities as high as 0.32 to 0.36 (Supplementary Figure 11), the average heritability of the combined PC for each imaging feature is modest. As the optimal heritability measures for high-dimensional multivariate measurements would be the mean heritability across PCs²³ (See Method), we found that the mean signal for NO, ND, and NF are 0.09 (95%CI 0.04 - 0.13), 0.06 (95%CI 0.02 - 0.10), and 0.05 (95%CI 0.01 - 0.09) respectively. Given the sum of the variance explained by the independent loci found to be significant, the discovered loci reached 59%, 60%, and 58% of the average SNP-heritabilities for NO, ND, and NF."

4. Line 156 – what is the evidence for a spatial gradient and what does it look like?

RESPONSE:

We agree with the reviewer that the discussion within the results section could include greater elaboration. We have moved that sentence to the discussion section and elaborated further on this point.

In the discussion section:

" Our results indicate widespread pleiotropies between the development of cortical surfaces and cerebral white matter. As the patterning of the mature brain is the end results of multiple molecular processes working from differentiation of neuroprogenitor cells, migration of neurons, to synaptic prunings⁶⁸, the relevant genes are unlikely to confine their effects on one single anatomically defined region. This is in line with findings from malformations of cortical development that the germline mutations of genes involved in cell migration would lead to global malformations instead of localized lesions⁶⁹. Many of the loci we discovered and replicated were also found to be associated with other imaging modalities across cortical and subcortical regions. We are not claiming that focal effects do not exist, indeed the variants we highlighted do have locally enriched signals. Instead we suggest that it is more likely that regional specification of the human brain is the ultimate result of a complex coordination of multiple distributed molecular processes⁷⁰ rather than that single genetic variants affect single anatomical regions. For instance, the group of genes belonging to CSPG had consistent associations with NO metrics. The association signals were enriched in multiple brain regions beyond previously reported ROIs^{18,21}, especially bilateral hippocampus. This astrocyte-dependent molecular

process may have more direct effects on the synaptic pruning in the hippocampal regions and then cascading down-stream to the associated fiber tracts."

5. Lines 201 – 213 – please explain what it means for a locus to be “druggable” and how this is determined.

RESPONSE:

Given the comments of both reviewers, we agreed that the druggable target analyses lends itself to misinterpretation. We have moved the entire section to the supplemental information and only mentioned the dataset overlaps in the main text, as the following:

In the last two sentence of the Abstract:

"Key molecular pathways involved in axonal growth, astrocyte-mediated neuroinflammation, and synaptogenesis during development were found to significantly impact the measured variations in tissue-specific imaging features. Our results shed new light on the biological determinants of brain tissue composition and their potential overlap with the genetic basis of neuropsychiatric disorders."

In the last two sentences of the Introduction:

"By investigating the spatial distribution of the associated effects, we highlighted critical molecular pathways involved in neuroinflammation and axonal growth, and the corresponding regions that may be susceptible to these processes. Signal overlap, at both the locus level and genome-wide, with neuropsychiatric outcomes indicate the functional relevance of our GWAS results, providing a foundation for further understanding of the biological underpinnings of neuropsychiatric disorders."

In the discussion of *VCAN* in the results section:

"VCAN, which encodes versican and is a lectican-binding chondroitin sulfate proteoglycan (CSPG), serves a critical role in astrocyte-mediated neuroinflammation⁴¹, and has potential interacting pharmacological targets^{42,43} (Figure 3d; Supplemental Information). ... Changes in the distribution of CSPGs in the hippocampal formation were observed among patients with schizophrenia and patients with bipolar disorder^{45,46}, linking our findings to neuropsychiatric outcomes."

In the discussion of *DPYSL5* in the results section:

*Our findings are also relevant to neuropsychiatric outcomes, as *CRMP* has been implicated in schizophrenia and mood disorders⁵⁰.*

6. Lines 235 – 253 – the associations with disorder results are weak and conclusions should be tempered accordingly.

RESPONSE:

Comparing to other shared loci analyses, the signals we found were relatively strong, indicating substantial pleiotropy between our imaging phenotypes and the neuropsychiatric outcomes. Nevertheless, to reduce the risk of overinterpretation, we soften the language on the interpretations of the correlations we found, as suggested by the reviewer as the following:

"While the limited resolution of LD blocks may contribute to this null finding, the evident similarities in the genome-wide level results may mean that pleiotropic effects, either horizontal or vertical, on neurodevelopmental traits are highly polygenic, sharing multiple loci but with different functional outputs. "

7. Line 251 – Pleiotropy is not demonstrated here. It is possible that the imaging measures mediate the link between genes and disease.

RESPONSE:

We appreciate the comments made by the reviewer regarding evidence of pleiotropy vs mediation. The pleiotropic effects can indeed be ‘vertical’ or ‘horizontal’. Vertical pleiotropy indicates that the imaging measures mediate the link between genes and diseases, while horizontal pleiotropy indicates that the loci were shared but the effects were independent. We did not differentiate between these two different types of pleiotropy when we mentioned this in our result section. We have revised the following paragraph in the results section to make this point clearer:

" While the limited resolution by the LD block can contribute to the lack of differences, the evident similarities at the genome-wide level may mean that the pleiotropic effects, either horizontal or vertical, on neurodevelopmental traits are highly polygenic, sharing multiple loci but with different functional outputs. "

Line 259 – this study does not really demonstrate the specific impact of molecular pathways – it simply interprets associations. No interventional studies are performed.

RESPONSE:

We understand the concerns of the reviewer and have made modifications accordingly. In particular, we also revised our title to reflect what we discovered were novel loci, not the molecular pathways.

Reviewer #3 (Remarks to the Author):

Manuscript Review

Multivariate genome-wide association study on tissue-sensitive diffusion metrics identifies key molecular pathways for axonal growth, synaptogenesis, and astrocyte-mediated neuroinflammation
Fan et al., 2021

Overview

Authors carried out GWAS for three novel neuroimaging phenotypes N0, ND and NF. N0 is most sensitive to anisotropically diffusing water in within cell bodies. ND is most sensitive to anisotropically diffusing water within oriented structures such as axons and dendrites. NF is a proxy measure of free water component. Authors carried out validation using polygenic voxel scores and carried out a series of downstream analysis to characterize the genomic loci discovered. Authors report that they identified key

molecular pathways involved in axonal growth, astrocyte-mediated neuroinflammation, and synaptogenesis during development. Additionally, they reported drug annotations of the primary GWAS findings for potential targets for pharmacological intervention

Overall impression

The paper addresses an important issue in neuroimaging genomics looking at diffusion measures. The phenotypes that the authors reported on would potentially advance the field in terms of how we understand connectivity in the brain. In addition, authors have used an interesting method that address pleiotropy across large number of neuroimaging measures to distill a composite GWAS of each connectivity-based neuroimaging phenotypes. Nonetheless, there are several queries that I have with regards to the approach to the results of the association analysis. In part, there is probably not enough details provided to the reader to fully appreciate the findings that the authors are reporting on. A minor point – if authors could format some of the Supplementary Tables in a spreadsheet format it would be very much more helpful to query some of the results that they are reporting on.

RESPONSE:

We sincerely thank the reviewer for positive comments and thorough considerations. We have updated the manuscript and the corresponding Supplemental Tables to make them more accessible to readers.

Specific Queries

1. Phenotypic Distributions and Descriptive Statistics

One of the novel aspects of the current report (and its strength) is in the neuroimaging phenotypes investigated. Would the authors be able to provide codes/scripts to demonstrate exactly how the phenotypes were derived from DTI measures? Related to that question, would authors also comment on the distribution of the derived phenotypes and how they related to the usual FA measures that are usually reported? Preferably, the results would have been more complete if the authors reported some descriptive statistics regarding their novel phenotypes N0, ND, NF and provided some visualization for the data.

RESPONSE:

We have shared the scripts to derive RSI measures based on multi-shell diffusion images on our publicly available Github page (See code availability section). To address the reviewer's specific query, the ND measure has similar spatial distributions as the more typically used FA measures, but has been shown to be better in tracking histological changes and brain development. Its advantage to the traditional FA measure may be due to the removal of noise driven by extra-cellular components. Furthermore, the RSI model is able to resolve crossing fibers, hence ND provides a more accurate measure of the degree of anisotropy than FA. To facilitate an intuitive understanding of these novel imaging features, we rendered the coronal slices of each imaging measure in Supplementary Figures 2 to 4, as well as their joint distribution in Supplementary Figure 5. We revised the result section accordingly as:

" N0, ND, and NF provide greater tissue specificity than the widely-used diffusion tensor metrics, have been useful in understanding variation of cellular organization within the human brain and are highly informative for human brain development ^{3,10,11,27}. The spatial distributions of these three tissue sensitive measures can be seen in Supplemental Figure 2-5. "

2. Data Analysis

Authors have indicated that the “multi-shell” method would be a more sensitive approach in understanding the biological underpinnings of diffusion neuroimaging measures. Authors report that the data analysis started with 100 x 100 x 130 voxels. It was then reported that the data was reduced to 5000PCs. I’m not quite sure how to reconcile the numbers reported in the methods where 50, 100, 500, and 1000 principal components were experimented with – I might have missed the conceptual decision to select 5000PCs. I guess the question comes up why 5000PCs - why not more or why not less? Were there simulations carried out to suggest that 5000PCs is the optimal number? Authors cited MOSTest as the “available code” but I’m not so sure if MOSTest was utilized in this context? That brings up another question, why did the authors decided to go with the CPC method as opposed to other methods?

RESPONSE:

The decision on using 5000 PCs is somewhat arbitrary. We did not go further because it already achieved more than 70% of total variance explained. Meanwhile, the numbers reported for 50, 100, 500, or 1000 PCs to be investigated further were used for the two component models of Fisher-combine proposed in the original CPC paper ²². Prior reports using both empirical data and simulations indicate the CPC is the best "practical" solution for high-dimensional data and the Fisher-combined inference can lead to better power ^{22, 29}. Therefore, we implemented the CPC while utilizing some of the core functions we used in the MOSTest code. To avoid potential confusion for readers, we have shared the CPC and utilized functions in a dedicated Github repository (See code availability section). We also revised both the result and method section to clarify the relationships between CPC and MOSTest as the following.

In the result section:

"For the discovery stage (UKB discovery set, imaging acquisition before 2019, n = 23,543), we used combined principal components (CPC) statistics ^{22,29} (Figure 1b) to identify associated loci from multivariate measurements. As a practical extension of our multivariate GWAS method MOSTest ²¹ package, CPC combines statistics from associations with finite number of principal components (PCs) and has close form expression on the null distribution without the need for permutations ²². Using the UKB discovery set, we calculated the PCs from the tissue feature across all voxels. From the whole-brain images in 2 mm resolution per voxel, spanning across 100 by 100 by 130 voxels, the first 5000 PCs were extracted and used in the subsequent analyses, explaining more than 70% of total variance of the imaging data (Supplementary Figure 6). Since all PCs are orthogonal to each other, the statistical inference can be based on combining the associations between genetic variants and each of the derived PCs (Figure 1b). Each of the PCs can be regarded as an orthogonal basis function with limited interpretability, yet the weighted combination of them can represent any spatial distribution (Supplementary Figure 7-9). CPC combined the association signals across PC for a given genetic variant and detect the genetic loci that are shared across multiple PCs, thus reducing the burden of multiple testing and the false detection on nuisance effects."

In the method section:

" Combined principal component GWAS (CPC)

In the present multivariate GWAS of RSI measures we implemented the CPC method ²² in the MOSTest package ²¹. When the covariance among input measures is identity, CPC testing statistic is mathematically equivalent to the MOSTest test. Therefore, the code base needed for performing CPC on

ultra-high dimensional imaging data is compatible to our MOSTest except the following two components: First, the imaging measures were undergoing eigen-decompositions to derive PCs. Second, the testing statistics were based on close-form solution instead of the permutation scheme. As we were working on identity covariance matrix with finite number of PCs instead of million of voxels, CPC is a practical alternative to the original MOSTest.

CPC has been shown to be a robust multivariate GWAS method that is well powered to detect loci across different scenarios^{19,22,29}. In our case, we optimized our power to detect genetic variants that shape the brain development, leaving traces in multiple brain regions. CPC enables the identification of loci that have association signals across multiple PCs, without the caveats of focusing on single brain regions. The procedures were as follows. First, the PCs and their corresponding eigenvectors were derived given the voxel-wise imaging data (Supplemental Figure 2-9). Each SNP was regressed on each of the derived PC scores, controlling for age, sex, 20 genetic PCs, genotyping batches, and intra-cranial volume. For a given SNP, the Wald statistics for each PC were combined as a simple linear sum (Figure 1b). Given that PCs are orthonormal, the sum of the squared Wald statistics follows the χ^2 distribution with k degrees of freedom for k PCs combined^{19,22}. Although several different combination functions can be used¹⁹, we found that the global-local combination with Fisher's method proposed in the original CPC paper has greatest power in detecting genetic loci²². Therefore, we experimented with four different global-local cut points (50, 100, 500, and 1000 PCs) to see which combinations yield the most discoveries. To reflect this experiment, we lowered the significance threshold to $p < 4.2e-9$ (corrected for 12 multiple comparisons, as 4 thresholds and 3 features were used in the current study)."

3. For the CPC approach, each SNP was regressed (with covariates) against each neuroimaging PC as phenotype. Which suggest for N0, ND, NF, 5000 GWASs were carried out, and the effect sizes were combined using the Wald statistics and df to generate a p-value for the association. The challenge with this type of pleiotropic methodology is that Wald statistics is not likely to have association direction. Just the magnitude. It would also mean that the method would not yield a heterogeneity measure, but in a way "collapse" both heterogeneity p-values and association p-values to give a much more powered form of association. If that's the case, it would be challenging to interpret the results of the "genetic overlap" even for other traits, or even fully show how each gene might be a potential "drug target" without the effect size for the variant association.

RESPONSE:

We are in complete agreement with the challenges of multivariate inference raised by the reviewer. How to adapt the analytic framework developed in the univariate context (such as genetic correlations with LDSC) to take in the multivariate inputs and how to interpret the multivariate results were critical hurdles to overcome. Fortunately, a recent publication with multivariate GWAS has laid the ground work for using commonly utilized softwares with multivariate statistics as input²³. The omnibus nature of the CPC picked up the signals in the spatial patterning regardless of how heterogeneous the effects were in the spatial domain, as long as the effects were not extremely sparse. Meanwhile, the replications in both UKB and ABCD ensured the effect sizes were consistent across independent datasets, i.e. with low heterogeneity across studies.

4. Results

Loci Discovery:

Authors report that a total of 503 unique loci across three of the neuroimaging measures in the report. Of these, 152 had not been previously reported. I don't see a table where these loci were specifically reported or annotated. These are probably in separate supplementary tables that the authors indicated (Supp Tables 2-7). It is recommended that authors present these novel loci in a separate table that are annotated so that readers could have a chance to evaluate the results of the findings. In addition, it would be good to have the source data for the non-novel loci reported in Figure 2d. Again, this would give the reader a chance to evaluate the findings presented in the report.

RESPONSE:

As the reviewer suggested, we added the information related to the novel/non-novel loci in the summary statistics for researchers to download (Supplementary Table 2-5). The added information includes both an indicator for novel/non-novel and the corresponding accession numbers if an overlap was found.

5. Validation:

Authors used a 'polyvoxel approach' to validate the findings of the discovery findings. This would involve summing across voxels weighted by the regression weight and eigenvector. I assume that coefficient k represents each voxel. How many voxels were included in this analysis? Relatedly, would authors see an association in their independent datasets without weighting on the regression coefficient – just using the eigenvector weights? And would the association effects/significance change after weighting – would the models be significant if compared against null $>$ eigenvector weighted only $>$ eigenvector + regression weights from discovery set.

RESPONSE:

Linear algebra can show that the polyvoxel score associations in the independent study are equivalent to the comparisons of the consistency in regression coefficients between the discovery set and validation set. Indeed, in our simulations, the sum of eigenvector weights would only be significant if the regression weights were monotonically scaled as the eigenvalues, which was not the case in the empirical analyses. Based on the theoretical framework, simulation results, and our recent publications on image scoring³³⁻²⁵, we only used the eigenvectors and regression weights derived from the discovery set to calculate the polyvoxel scores in the replication sets, allowing us to examine the generalizability of our findings.

6. Authors also attempted to examine enrichment in brain regions using the polyvoxel approach. The regions were included in Supplementary Table 11 (not Supplementary 10 as indicated in the manuscript). There was no further discussion about this analysis in the results.

RESPONSE:

Due to the large number of discovered loci, we only highlighted the results of regional enrichment analyses for loci we discovered and replicated. To facilitate further investigations on other loci we identified, we placed all the results of the regional enrichment analyses in the Supplementary Tables 6-8.

7. Code/Data Availability

Authors cited MosTest and ABCD pipelines as available code. However. It would be critical for some of the computational pipelines for the polyvoxel approach and estimation of N0, ND, and NF to be available a well since these are the aspects that make the report novel.

RESPONSE:

We agree with the reviewer's comments and have now put all the relevant codes and scripts in a dedicated github repository for reference (See code availability section in the revised main text).

REVIEWER COMMENTS

Reviewer #1 (Remarks to the Author):

The authors have made substantial and effective efforts to address all my comments on initial review.

Reviewer #2 (Remarks to the Author):

All comments have been addressed.

Reviewer #3 (Remarks to the Author):

I would like to thank the authors for their response. I am happy for the article to move forward.

RESPONSE TO REVIEWERS

Reviewer #1 (Remarks to the Author):

The authors have made substantial and effective efforts to address all my comments on initial review.

Reviewer #2 (Remarks to the Author):

All comments have been addressed.

Reviewer #3 (Remarks to the Author):

I would like to thank the authors for their response. I am happy for the article to move forward.

RESPONSE:

We are delighted that the reviewers found our revisions satisfactory. We sincerely thank all the reviewers for their insightful comments and suggestions.